# Reliability of a Risk-Factor Questionnaire for Osteoporosis: A Primary Care Survey Study with Dual Energy X-ray Absorptiometry Ground Truth

**DOI:** 10.3390/ijerph18031136

**Published:** 2021-01-28

**Authors:** Maria Radeva, Dorothee Predel, Sven Winzler, Ulf Teichgräber, Alexander Pfeil, Ansgar Malich, Ismini Papageorgiou

**Affiliations:** 1Institute of Diagnostic and Interventional Radiology, Jena University Hospital—Friedrich Schiller University Jena, Am Klinikum 1, 07747 Jena, Germany; maria_radeva@yahoo.de (M.R.); ulf.teichgraeber@med.uni-jena.de (U.T.); 2Institute of Radiology, Suedharz Hospital Nordhausen, Dr.-Robert-Koch-Str. 39, 99734 Nordhausen, Germany; dteepredel@t-online.de (D.P.); sven.winzler@shk-ndh.de (S.W.); ansgar.malich@shk-ndh.de (A.M.); 3Department of Internal Medicine III, Jena University Hospital—Friedrich Schiller University Jena, Am Klinikum 1, 07747 Jena, Germany; alexander.pfeil@med.uni-jena.de

**Keywords:** osteoporosis, survey, bias, FRAX, patient management, self-awareness

## Abstract

(1) Purpose: Predisposing factors to osteoporosis (OP) as well as dual-source x-ray densitometry (DXA) steer therapeutic decisions by determining the FRAX index. This study examines the reliability of a standard risk factor questionnaire in OP-screening. (2) Methods: *n* = 553 eligible questionnaires encompassed 24 OP-predisposing factors. Reliability was assessed using DXA as a gold standard. Multiple logistic regression and Spearman’s correlations, as well as the confounding influence of age and body mass index, were analyzed in SPSS (IBM Corporation, Armonk, NY, USA). (3) Results: Our study revealed low patient self-awareness regarding OP and its risk factors. One out of every four patients reported a positive history for osteoporosis not confirmed by DXA. The extraordinarily high incidence of rheumatoid arthritis and thyroid disorders likely reflect confusion with other diseases or health anxiety. FRAX-determining risk factors such as malnutrition, liver insufficiency, prior fracture without trauma, and glucocorticoid therapy did not correlate with increased OP incidence, altogether demonstrating how inaccurate survey information could influence therapeutic decisions on osteoporosis. (4) Conclusions: Contradictive results and a low level of patient self-awareness suggest a high degree of uncertainty and low reliability of the current OP risk factor survey.

## 1. Introduction

The World Health Organization (WHO) and the International Society for Clinical Densitometry (ISCD) define osteoporosis (OP) as the reduction of the bone mineral density (BMD) by equal or more than 2.5 standard deviations from the average of a healthy young adult pool, matched for the biological sex and ethnicity (T-score ≤ −2.5) [1,2,3,4]. Precise diagnosis is of major concern; thus, many studies revolve around the problem of over-or underestimation [5], its associated fracture morbidity, and costs [6,7,8]. Despite the wide variety of BMD-estimation methods, the dual-energy x-ray absorptiometry (DXA) is the only WHO-qualified approach [1,3,4,9,10]. According to the criteria by the National Osteoporosis Foundation (NOF), OP is defined by the minimum T-score at any measuring site [3,11,12,13].

OP is associated with increased fracture risk and requires therapeutic intervention. The FRAX^®^ index is a WHO-endorsed fracture probability prediction model [1,14], which implements the femoral neck BMD but also various risk factors such as previous atraumatic fracture, family history, use of glucocorticoids, and rheumatoid arthritis (RA) to predict the 10-year risk for a major osteoporotic fracture or an osteoporotic hip fracture. FRAX is the current gold standard for OP-treatment initiation. Judicious therapy administration determines the therapeutic benefit [15,16,17,18,19] and prevents considerable side effects [14,20,21]. The NOF treatment guidelines [3] accept a FRAX score of >3% as a threshold for therapy initiation for an osteoporotic hip fracture or a FRAX score >20% for a major osteoporotic fracture, which means that the therapeutic decision is dependent not only on the BMD measurement but also on the risk factor profile.

Patient history, predisposing clinical factors, and morphometric data were collected in advance using an adapted questionnaire [3,22] to calculate the FRAX index [23], i.e., the 10-year fracture risk that steers treatment decisions. The questionnaire was filled out without physician consultancy as part of the OP-screening. Our study aimed to define the reliability of the risk-factors questionnaires collected in our OP outpatient clinic, accounting for *n* = 553 in 3 years.

In summary, our results elucidated low questionnaire reliability, reflecting a low patient self-awareness regarding both the diagnosis of OP and its associated risk factors. Erroneous anamnestic information is a highly-ranked limiting factor in the survey’s predictive value and has high therapeutic relevance for calculating the FRAX index.

## 2. Materials and Methods

### 2.1. Study Design and Participant Flow

We conducted a retrospective survey study with purposive sampling of all patients investigated in our center between 6/2016 and 6/2019. The study design was based on established standards for survey data collection, analysis, and reporting [24,25]. Participant flow is graphically illustrated in Figure 1. All patients filled in a survey with morphometric and risk-factor information (Appendix A) along with the written consent as a prerequisite for the DXA examination without a physician’s consultancy. From a patient registry of *n* = 560 surveys, *n* = 7 subjects were rejected due to insufficient identification information. No gender, age, or risk factor bias could be claimed for the excluded questionnaires. No other exclusion criteria were applied. The eligible database consisted of *n* = 553 (478 females).

### 2.2. Questionnaire Design and Questionnaire Filling

The questionnaire (Appendix A) was designed by AM based on the clinical factors predisposing people to osteoporosis as defined by the NOF criteria [3] as well as the recommendations of the American College of Radiology (ACR), the ISCD, and the International Atomic Energy Agency (IAEA) [11,26]. Apart from the FRAX-determining factors (previous fracture, OP family history, smoking, glucocorticosteroids, rheumatoid arthritis, secondary osteoporosis, and alcohol consumption), the questionnaire of this study was enriched in risk factors statistically correlated with osteoporosis such as eating disorders and hormonal dysfunction [27]. The survey was conducted face-to-face between the patient and the radiographer as part of the informed consent before the examination. The questions were piloted with 20 subjects and tested for clarity and validity by three physicians (AM, AP, IP), one of whom has expertise in osteology (AP). The questions had a closed format with binary answers, which ensured clarity of interpretation. The questionnaire was handed out in a DIV-A4 printed hard copy. The questions were numbered and grouped in topics without randomization [24,25]. A question on osteoporosis history served as positive quality control of patient self-awareness. An example of the questionnaire is provided in the Appendix A.

### 2.3. Dual-Source X-ray Absorptiometry Imaging, Reference Database, and T-Score Interpretation

DXA imaging was performed in a Hologic Discovery Wi (S/N83214) V. 13.6.0.2 linear, dual-energy (100 kV and 140 kV) X-ray fan-beam scanner (Hologic Inc., Bedford, MA, USA) utilizing a motorized table, c-arm, and multi-element detector array. The device’s operation mode and quality control adhered to the manufacturer’s recommendations for operation and maintenance, conforming to the technical recommendations of the ISCD and IAEA [11,26]. Standardized measurements included (i) the femur neck, (ii) the total hip, which is the average of the femoral neck and intertrochanter measurements, (iii) the lumbar spine, which is the average of the L1–L4, (iv) and the distal forearm (or 33% of the radius). Lateral spine views were not implemented for diagnostic purposes in accordance with the ISCD official positions [2,13]. Incorporation of barium, iodinated, or gadolinium-based contrast enhancers was prohibited two weeks ahead of the DXA; calcium tablets were ceased 24 h before the examination. All patients were examined in a hospital gown to minimize the effect of clothing. For the PA-spine (posteroanterior view of the lumbar spine) examination, the lumbar lordosis was alleviated with a manufacturer-approved positioning block under the knees and the pelvis. The hip was examined while abducted in inner rotation and immobilized using a positioning device approved by the manufacturer.

The reference database for the Caucasian bone mineral density (BMD in g/cm^2^) was provided by the manufacturer (Hologic) and fulfils the WHO and ISCD recommendations [2,28]. The database is a built-in software feature without an interface for user manipulations.

The T-score (T-score = standard deviations of the measured BMD from the average of a sex-matched reference database) and the Z-score (Z-score = standard deviations of the measured BMD in g/cm^2^ from the average of a sex-and-age-matched reference database) were calculated using the manufacturer’s standards. The examination report contains the BMD, T-scores, and Z-scores for each measuring site and the average scores for the whole area. The literature on the technical aspects and clinical applications of the T- and Z-scores was excellently reviewed by the Human Health Series of IAEA [29].

According to the recent positions of the ISCD, the WHO, and the International Osteoporosis Foundation (IOF) on the diagnosis and treatment of osteoporosis [1,2,13], the standard DXA protocol included a measurement of the left hip and a posteroanterior view of the lumbar spine (PA-spine) from the first to the fourth lumbar vertebrae (L1–L4). In cases where endoprosthesis material was present in the left hip, the right hip was used instead. Cemented, stabilized, or heavily degenerated lumbar segments were excluded from the DXA measurement. A lumbar spine measurement was considered valid with at least two valid spines [12]. In case of necessary exclusion of the lumbar spine or both hips, and in patients with hyperparathyroidism, an additional measurement of the distal forearm (33% of the radius) was implemented [1,2,13,26].

Two radiologists validated the DXA quality: one with intermediate experience and a board-certified radiologist with more than 15 years of experience in DXA interpretation (IP or DP, and AM).

### 2.4. Ground Truth

The T-score threshold for osteoporosis was ≤−2.5 at any measuring site, as defined by the WHO criteria [1,2]. For testing the effect of the measurement site on the OP diagnosis depending on the risk factor profile, we analyzed the database in repetition, implementing two alternative measuring sites; (i) the total hip T-score, and (ii) the minimum T-score between hip, femur neck, PA-spine, and radius, which is the current WHO standard [1,11,12,26,30,31,32,33,34].

### 2.5. Statistical Analysis and Handling of Missing Data

Logistics regressions and descriptive statistical data processing were performed with the Microsoft Office 365 suite (Microsoft Corporation, Redmond WA, USA). The study was powered to 95% (a = 0.05) for an odds ratio of 1.8 (required *n* = 245) using G*Power 3.1.9.7 [35,36]. The independent influence of each risk factor on the occurrence of OP was expressed in linear regressions using Spearman’s method. The combined effect of risk factors was analyzed using a multiple logistic regression analysis with a Wald-test. The validity of multiple logistic regression was confirmed with a Homer–Lemenshow test when *p* >> 0.05. Regression analysis and histogram plotting were performed with the SPSS (IBM GmbH, Ehningen, Germany). Graphical editing was accomplished using the open source platform Inkscape (GPL v2+, https://inkscape.org). Percentages were rounded up to two decimal places, statistical values are reported with an accuracy of four decimal digits.

### 2.6. Ethical Statement

The patient database was derived from a single center. A generic or individual indication justified medical radiation exposure. Data were analyzed retrospectively and fully anonymized according to the ethical standards laid down in the 1964 Declaration of Helsinki and its amendments, the European Regulation 536/2014, and the Good Clinical and Scientific Practice protocols of the hosting institution. All patients were informed about the safety of ionizing radiation and radiation protection. The Institutional Review Board approved the study (protocol number 2019–1450) and waived the mandate of obtaining legally effective informed consent from the included subjects.

## 3. Results

### 3.1. Response Rate, Dealing with Missing Data, and Profile of Non-Responders

The average age for total/male/female was 67.78 ± 12.28/68.44 ± 13.37/67.68 ± 12.10 (average ± standard deviation) years old (y.o.), and the dominant age group for both genders was 75–80 years old (Figure 2a,b). The number of complete questionnaires was *n* = 211; incomplete questionnaires were analyzed on a single-question basis with linear regression but excluded from the multiple logistic regression workflow (Table 1).

The single-question response rate of *n* = 553 eligible patients (478 females) was more than 90% (Appendix A). A total of *n* = 211 (174 females) completed the full questionnaire. Responses from incomplete questionnaires were processed with Spearman’s univariate analysis and were excluded from the multivariate analysis. The *n* for each question’s univariate analysis (total, female) is provided in Appendix A.

We analyzed the sex- and age-profile of non-responders for each question numbered from Q1 to Q24 to exclude intrinsic bias (Appendix A). The *t*-test between the age histograms of responders and non-responders showed no significant difference (*p* > 0.05), which excluded a possible age-dependent bias in the logistic regression.

### 3.2. The Osteoporosis Incidence Is Affected by the Bone Mineral Density Measuring Site

The incidence of osteoporosis in the tested population was (f/m) 9% and 5% according to the hip score and 34/24% based on the minimum T-score (Figure 2c,d). The discrepancy between hip and minimum T-score was statistically significant, Mc-Nemar’s test *p* < 0.001. At the level of a single individual, Cohen’s analysis showed a fair interobserver agreement between hip and minimum T-score of 76.13% with a Cohen’s kappa of 0.33. Univariate and multiple logistic regressions were performed for the total sample and females separately (Table 1). Due to the low sample size of male subjects and the low incidence of osteoporosis in males, our database was not adequately powered for a reliable multiple logistic regression in the male population (Homer–Lemenshow test *p* < 0.05).

### 3.3. Assessment of the Patient’s Awareness Regarding Osteoporosis

To assess the self-awareness of tested individuals regarding osteoporosis, patients were questioned about their history of osteoporosis (OP) prior to the examination (Table 1, Q12). Of the 553 patients questioned, *n* = 239 (51.18%) reported a positive OP-history. However, DXA revealed that only *n* = 180/553 (32.55%) had osteoporosis according to the criterion of minimum T-score. A false positive rate (FPR) of 0.23 means that approximately one every four patients had low self-awareness or an insufficient understanding of osteoporosis [37]. Unsatisfactory communication between patients and general practitioners [38] might be the reason, and highlights the need for a physician’s consultancy to assess the risk factor profile accurately. Questionnaires with false-positive OP-history responses were considered eligible because this study aimed to encompass the realistic clinical situation, including an inevitable percent of vague answers.

### 3.4. Association between Risk Factor Survey and DXA as Ground Truth

Eating disorders (Table 1, Q6). The question about eating disorders (ED), such as anorexia nervosa and bulimia, was answered by *n* = 545/469 (total/female). Positive history for ED was accounted for in *n* = 19/17 (total/female), accounting for 4.05/3.63% of the responders, respectively. ED significantly correlated with an increased osteoporosis occurrence regardless of the measuring site according to multiple logistic regression, *p* < 0.01. The odds ratio (OR) for the hip measurement was 20 times higher than the minimum T-score for both the total sample and females (Table 1).

Table 1 shows the correlation of risk factors with an increased incidence of osteoporosis. Linear regression used the Spearman’s method, multiple logistic regression with Homer–Lemeshow test, and Ward’s statistics. Table 1 includes only the parameters showing a significant negative or positive correlation with osteoporosis. For the complete statistical report, including the parameters without a significant correlation to osteoporosis, please refer to the Appendix A. Statistically significant results are bold-enhanced and italic. 

In addition to the correlation of ED with OP, we examined the influence of the body mass index (BMI) on BMD. Linear regression analysis revealed a weak correlation between ED and pathologically reduced BMI, with most ED patients being in the range of normal weight or obese (Figure 3). The BMI of individuals claiming eating disorders was significantly lower compared to that of the non-ED fraction (*p* < 0.05, Spearman’s correlation), but varied nevertheless from underweight to overweight (Figure 3b). The BMI of non-responders (n.r.) to the ED-question (Figure 3b) was derived from the histogram peak; hence it is unlikely to create a bias. The Spearman analysis between BMI and BMD showed the expected positive correlation between the two parameters, *p* << 0.001 with correlation coefficients of R = 0.301/0.303 for femur neck and minimum T-score, and R = 0.375/0.342 for total hip and spine BMD.

All in all, ED and lower BMI are independently correlated with OP, as the vast majority of the ED group was of normal weight or overweight. The DXA contamination by the lean body fat at high BMI groups complexifies the ED-OP interpretation. In general, the higher incidence of OP in the ED group cannot be explained exclusively by the reduced body weight [39,40].

Sex hormone disorders (Table 1, Q21). The question of sex hormone disorders (SexHorm) was answered by *n* = 477/414 (total/female). Positive history for SexHorm was accounted for in *n* = 58/53 (total/female), accounting for 12.16/12.80% of the responders. In a multiple logistic regression, the association of SexHorm with the hip T-score revealed a *p* = 0.02/0.03 with an OR = 0.05/0.01 in total/females, and with the minimum T-score a *p* = 0.003/0.008 with an OR = 0.14/0.11. Hence, SexHorm had a significant inverse correlation with OP occurrence regardless of the measuring site, meaning that OP was more likely to occur in the group not suffering from sex hormone disorders. This counterintuitive result is strongly biased by low self-awareness on the nature of sex hormone problems. It should be noted that the questionnaire was not corrected for hormonal substitution therapy, which can be an additional confounding factor.

Rheumatoid arthritis (Table 1, Q2). The question regarding rheumatoid arthritis (RA) was answered by *n* = 523/450 (total/female). A positive history for RA was found in *n* = 103/86 (total/female), accounting for 19.69/19.11% of the responders. The extraordinarily high RA incidence signalizes that the RA is not distinguished from osteoarthritis or other unspecific arthritic changes [41,42], which likely explains the negative correlation between RA and OP (minimum T-score) in the multiple logistic regression, *p* = 0.05/0.01 OR 0.34/0.09 for total/females, respectively. The hip T-score was not correlated with a positive RA response either, *p* = 0.51/0.67 (fem/total) univariate, and *p* = 0.46/1.00 multivariate analysis (Table 1). Hence, despite the proven correlation of RA with OP [43,44], our paradigm shows that patient uncertainty regarding the nature of joint disease imposes a tremendous influence for the prognosis of osteoporotic fractures using the FRAX index.

Thyroid and parathyroid hormonal disorders (Table 1, Q20). The question of hormonal disorders of the thyroid or parathyroid glands (ThPTh) was answered by *n* = 533/457 (total/female). Positive history for ThPTh was accounted for in *n* = 164/150 (total/female), accounting for 30.77/32.82% of the responders. This percentage is extremely high, considering the expected prevalence of approximately 2% for thyroid disease [45] and an even lower prevalence by a factor of 100 for parathyroid diseases in European countries [46]. Correction for the age group could not explain the increased reported thyroid disease incidence either. This discrepancy most likely reflects a low self-awareness regarding hormonal diseases since no bias factors or population selection criteria influenced our study. ThPTh revealed a significant correlation only with the hip (*p* = 0.02, OR = 4.23 for total, *p* > 0.05 for females) but not with the minimum T-score (*p* > 0.05 for total and females) in the multiple logistic regression.

Anticoagulation medication (Table 1, Q22). The question about anticoagulation therapy (Anticoag) was answered by *n* = 536/462 (total/female). Positive history for Anticoag was encountered in *n* = 158/124 (total/female), i.e., 29.48/26.84% of the responders. Anticoag was not found to be an independent effector of the OP incidence (*p* >> 0.05, multiple logistic regression to minimum T-score). However, a significant correlation occurred when analyzing for the hip score, *p* = 0.007 OR = 6.05 multiple logistic regression for the total sample but not for the females (*p* = 0.149, female, multiple logistic regression to hip T-score).

Cancer (Table 1, Q23). The question regarding cancer history (CA) as a risk factor was answered by *n* = 536/462 (total/female). A positive history for CA was encountered in *n* = 87/79 (total/female), 16.23/17.10% of the responders. As an independent factor, CA was not significantly correlated with OP (*p* > 0.05 for hip and minimum T-score). This counterintuitive result might rely on the short-term history from the CA diagnosis in our patient collective. DXA screening is a standard of care before launching antihormone therapy for breast cancer in our center. Multiple logistic regression unravels the expected CA correlation with OP when implementing the hip T-score as a dependent variable (*p* = 0.02/0.007 and OR = 8.36/29.16 for total/female). The correlation is surprisingly weaker and significant only for the females when implementing the minimum T-score (*p* = 0.08/0.018 and OR = 2.66/4.6 total/female).

Summarizing the above, clinical conditions such as Anticoag and CA are stronger effectors of the hip T-score than they are for the minimum T-score, which is the current diagnostic standard for OP. This finding highlights the significance of a critical and individualized interpretation of DXA results with the risk factor profile, as suggested by the ACR guidelines 2018.

## 4. Discussion

This study aimed to improve OP screening and patient management by defining the risk factor questionnaire’s reliability. According to the DXA ground truth and the results of the multivariate analysis, from a list of 24 questioned risk factors, two were significantly related to an increased incidence of OP: eating disorders (OR 10.36) and cancer history (OR 4.6). Anticoagulation therapy (OR 6.05) was associated with a pathological total hip T-score but not with OP. The remaining predisposing factors (Appendix A) did not correlate with an increased OP-incidence. Surprisingly, OP occurrence was indifferent to FRAX-relevant parameters such as a fracture history, positive family history, and glucocorticoid treatment. Relevant diseases such as rheumatoid arthritis (OR 0.34) and sex hormone dysfunction (OR 0.14) even showed a negative correlation to OP. Our results suggest that a low level of patient awareness and uncertainty regarding OP and its associated risk factors strongly influences therapeutic decisions regarding osteoporosis via the FRAX index.

This study is directly intercalated to the quality of daily clinical management. Even in financially strong, adequately equipped radiological departments, the standard-of-care osteological imaging is performed by radiographers with remote medical supervision. A questionnaire including FRAX-related factors, such as the ones tested in this study, is filled out by the patients without one-to-one medical support. The risk factors are subsequently fed in the FRAX-calculator to estimate the FRAX-score and determine the need-to-treat according to the 10-year fracture risk (www.sheffield.ac.uk/FRAX/tool). The beneficial implications of obtaining a valid patient questionnaire are obvious since anamnestic accuracy influences the treatment decision of osteoporosis. Moreover, the accuracy and reliability of the risk factor survey is vital for high-risk patient selection, especially in non-DXA screening protocols, such as the OP-screening with a questionnaire and quantitative ultrasound [47] or by using the Osteoporosis Screening Tool (OST) [48,49]. Evidence-based screening for OP was attempted by the Risk-Stratified Osteoporosis Strategy Evaluation (ROSE) study [50,51], which implemented a two-step process with survey screening and DXA screening of a selected population. The current study reveals that the implementation of questionnaires as a standalone or first-line screening method calls for cautious data interpretation and requires critical medical supervision to provide reliable results.

No conflicts of interest or sampling bias influenced the reliability and credibility of the current research findings. A great effort was put into analyzing the profile of non-responders and eliminating the age bias to each question individually. The low self-awareness regarding OP, reflected by an excess in positive responses to OP history, has puzzled the osteological community before [37,52], and partly reflects an unsatisfactory communication between patient and general practitioner [39] as well as possible influences from social media and health anxiety. Individual, personalized announcement of the DXA results is suggested as a best practice [38]. Although we did not collect data on previous OP-treatment, our center is the sole recruiting DXA-unit in the region, with a high rate of long-term patient follow-up. Hence, the discrepancy is not likely to reflect an effective prior therapy of existing osteoporosis.

Study weaknesses might have affected the outcome of some questions. A long list of proven OP-related risk factors did not reveal a positive correlation in our study, putting in question the validity of patient surveys (Appendix A) [50]. Nonetheless, the limited number of samples hampered some possible correlations observed from the survey, such as the malabsorption syndrome (9/485 positive/total responses), developmental (6/543) and growth retardation (4/537), organ transplantation (8/524), liver insufficiency (18/538), and anticonvulsive therapy (16/529). Positive correlations of those factors with osteoporosis are competently reviewed in high reputation specialty books [27,53]. Chronic gastrointestinal diseases (44/544) and renal insufficiency (44/538 positive response rate) did not correlate with OP despite being adequately represented in our sample. It is worth commenting that this questionnaire specified neither the type of gastrointestinal disease (gastritis, inflammatory bowel disease, lactose intolerance, or unclassified patient complaints) nor the degree of renal insufficiency. Premature (86/462) or late menopause (36/445 positive rate, respectively) did not correlate with an increased OP incidence despite adequate representation. In both cases, hormonal supplementation was not clarified. Height loss of more than 4 cm (183/525), long immobilization periods (96/509), spontaneous fracture in the absence of trauma (121/515 positive response rate in the total sample), family history of OP (209/505), fracture without major trauma (121/515), and the use of glucocorticoids (64/526 positive response rate) did not correlate with an increased OP incidence regardless of the measuring site. The clinical conditions listed above are highly relevant to the occurrence of osteoporotic fractures [54,55,56] and crucial for the calculation of the FRAX^®^ index for hip and general osteoporotic fracture risk assessment [1,3,14,57,58,59]. The FRAX^®^ index [60] is the current gold standard, endorsed by the WHO (https://www.sheffield.ac.uk/FRAX/index.aspx) as well as by the NOF therapeutic recommendations on OP [3]. Hence, low self-awareness and imprecise risk factor information could directly impact therapeutic decisions.

## 5. Conclusions

The NOF-questionnaire is a determinator of the FRAX-index and tunes therapeutic decisions. This study tested the reliability of an extended NOF-questionnaire version as applied in the osteological outpatient clinic. Summarizing the findings from 24 questions about OP-related risk factors, only eating disorders (OR 10.36) and cancer (OR 4.6) were associated with an increased OP incidence. Many established OP risk factors, including numerous FRAX-estimates, were not associated with the OP-occurrence in our study. Low self-awareness and patient uncertainty are considered major confounding factors. This result raises doubt in the reliability of patient surveys as stand-alone tools for the prediction of OP. We believe that a targeted physician–patient interaction and reliable risk factor estimation are necessary areas of improvement in outpatient osteoporosis care. 

## Figures and Tables

**Figure 1 ijerph-18-01136-f001:**
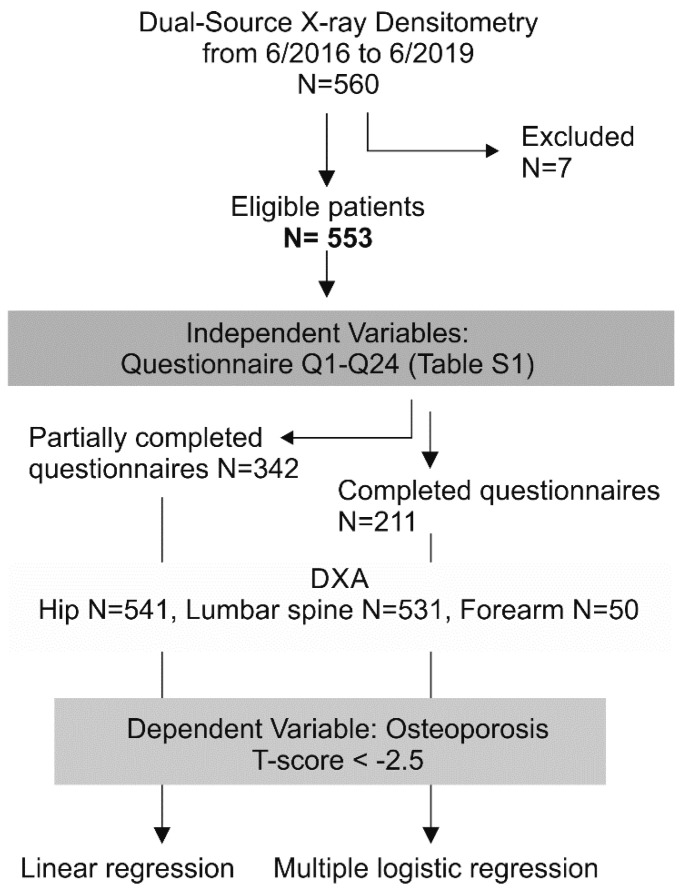
Flowchart of patient flow and study design.

**Figure 2 ijerph-18-01136-f002:**
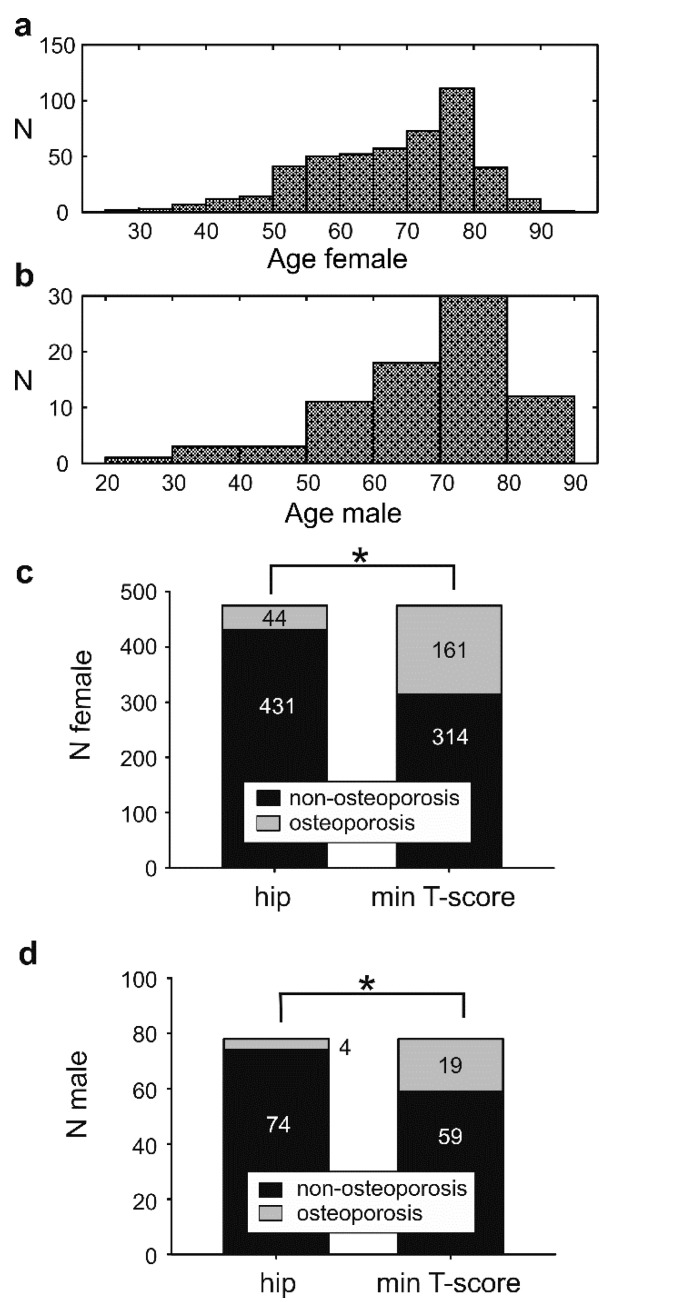
Age distribution and incidence of osteoporosis in the sample. (**a**) Age distribution histogram of the female subjects, (mean ± standard deviation) 67.68 ± 12.10 y.o. (years old); (**b**) age distribution of the male subjects, 68.44 ± 13.37 y.o.; (**c**) incidence of osteoporosis (OP) in female subjects was 9.3% using as a criterion the total hip T-score and 33.89% using the minimum T-score, according to the World Health Organization recommendation; (**d**) the incidence of OP in the males was 5.1% using as a criterion the total hip T-score and 24.6% using the minimum T-score. The interobserver agreement between the hip and minimum T-score for both sexes was 76.13%, Cohen’s kappa 0.33, *p* < 0.001 McNemar’s test. Significant statistical difference at *p* < 0.05 is denoted with an asterisk.

**Figure 3 ijerph-18-01136-f003:**
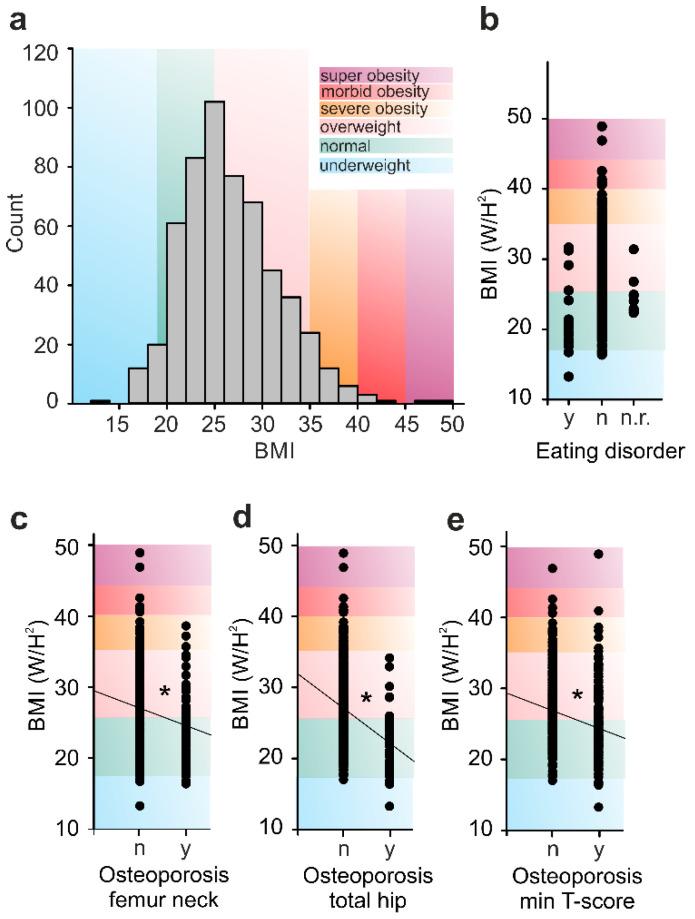
Correlation of the body mass index (BMI) with osteoporosis (OP). (**a**) Histogram of the BMI showing a right-skewed distribution, with more than 50% of the sample being overweight to morbidly obese. (**b**) A scatterplot showing how the BMI correlates with eating disorders (ED), yes = y, no = n, n.r. = no response. The BMI of patients with an ED was lower than that of patients without ED, however, all were not necessarily underweight, *p* < 0.05 Spearman’s correlation. (**c**–**e**) BMI correlation with the OP in (**c**) the femur neck r_s_ −0.177, (d) total hip r_s_ −0.273, and (**e**) based on the minimum T-score r_s_ −0.197. Regardless of the measuring site, patients with OP had a lower BMI than that of non-OP patients, *p* < 0.001 Spearman’s correlation. The correlation coefficient of Spearman (r_s_) is indicated by a solid diagonal line. Statistically significance at *p* < 0.05 is denoted with an asterisk.

**Table 1 ijerph-18-01136-t001:** Risk factor correlation with increased osteoporosis incidence.

*n* Total (Male/Female) = 553 (78/475), Age (68 ± 12/68 ± 13) y.o.	Ground Truth Hip, Index Osteoporosis										Ground Truth Min T Score, Index Osteoporosis							
Survey Question	Univariate Analysis, Linear Regression (Spearman)				Multiple Logistic Regression, Feed All 24 Covariates						Univariate Analysis, Linear Regression (Spearman)			Multiple Logistic Regression				
					Hosmer-Lemeshow Statistic ALL: 2.700 (*p* = 0.952) Hosmer-Lemeshow Statistic FEM: 0.626 (*p* = 1.000)									Hosmer-Lemeshow Statistic ALL: 9.345 (*p* = 0.314) Hosmer-Lemeshow Statistic: 3.484 (*p* = 0.900)				
	*n*	*P*	*R*	*R* ^2^	*n*	Coefficient	*SE*	Wald Statistic, Chi-Squared	*p*	Odds Ratio	*p*	*R*	*R* ^2^	Coefficient	*SE*	Wald Statistic, Chi-Squared	*p*	Odds Ratio
Q2 RA total	523	0.5100	0.0288	0.0008	211	−0.6370	0.8720	0.5340	0.4650	0.5290	0.2550	0.4990	0.0025	***−1.0730***	***0.5480***	***3.8370***	***0.0500***	***0.3420***
Q2 RA female	450	0.6730	0.0199	0.0004	174	−19.4040	>1000	0.0000	0.9970	0.0000	0.5090	0.0312	0.0010	***−2.4340***	***0.9780***	***6.1920***	***0.0130***	***0.0877***
Q6 ED total	***545***	***<0.001***	***0.1970***	***0.0390***		***5.2650***	***1.4960***	***12.3810***	***<0.001***	***193.5130***	0.1530	0.0163	0.0038	***2.3380***	***1.0700***	***4.7760***	***0.0290***	***10.3600***
Q6 ED female	***469***	***<0.001***	***0.2190***	***0.0479***		***6.2800***	***2.3800***	***6.9600***	***0.0080***	***533.9250***	0.0870	0.0790	0.0062	***2.9230***	***1.2900***	***5.1330***	***0.0230***	***18.5980***
Q12 OP total	***504***	***<0.001***	***0.1620***	***0.0264***		***1.4460***	***0.7150***	***4.0870***	***0.0430***	***4.2440***	***<0.001***	***0.3570***	***0.1270***	***1.9030***	***0.4190***	***20.6720***	***<0.001***	***6.7060***
Q12 OP female	***436***	***0.0010***	***0.1570***	***0.0246***		1.4200	0.9580	2.1980	0.1380	4.1390	***<0.001***	***0.3550***	***0.1260***	***1.9340***	***0.5020***	***14.8590***	***<0.001***	***6.9190***
Q13 BD total	467	0.1030	0.0715	0.0057		0.6060	0.7340	0.6810	0.4090	1.8330	***<0.001***	***0.2013***	***0.0454***	−0.0888	0.4760	0.0348	0.8520	0.9150
Q13 BD female	***407***	***0.0030***	***0.0735***	***0.0054***		0.9270	1.0290	0.8120	0.3680	2.5270	***<0.001***	***0.2950***	***0.4640***	−0.2090	0.6130	0.1170	0.7330	0.8110
Q20 ThPTh total	533	0.1990	0.0557	0.0031		***1.4490***	***0.6220***	***5.4330***	***0.0200***	***4.2580***	0.1560	0.0615	0.0038	0.0997	0.3800	0.0688	0.7930	1.1050
Q20 ThPTh female	457	0.1470	0.0680	0.0046		0.7340	0.9110	0.6490	0.4210	2.0830	0.3490	0.0439	0.0019	−0.1850	0.4660	0.1570	0.6920	0.8310
Q21 SexHorm total	***477***	***0.0420***	***0.0930***	***0.0085***		***−3.1020***	***1.3640***	***5.1720***	***0.0230***	***0.0450***	***0.0210***	***0.1060***	***0.0112***	***−1.9950***	***0.6810***	***8.5820***	***0.0030***	***0.1360***
Q21 SexHorm female	***414***	***0.0400***	***0.1010***	***0.0102***		***−4.3590***	***1.9880***	***4.8060***	***0.0280***	***0.0128***	***0.0150***	***0.1200***	***0.0144***	***−2.1990***	***0.8360***	***6.9260***	***0.0080***	***0.1110***
Q22 Anticoag total	536	0.1080	0.0695	0.0048		***1.8000***	***0.6630***	***7.3690***	***0.0070***	***6.0470***	0.9330	0.0036	0.0000	0.0989	0.4060	0.0595	0.8070	1.1040
Q22 Anticoag female	462	0.1350	0.0697	0.0049		1.2960	0.8970	2.0870	0.1490	3.6550	0.9280	0.0042	0.0000	−0.2680	0.5180	0.2670	0.6060	0.7650
Q23 CA total	536	0.2770	0.0470	0.0022		***2.1240***	***0.9040***	***5.5170***	***0.0190***	***8.3610***	0.3960	0.0367	0.0014	0.9810	0.5660	3.0010	0.0830	2.6670
Q23 CA female	462	0.3180	0.0466	0.0022		***3.3730***	***1.2560***	***7.2100***	***0.0070***	***29.1600***	0.4590	0.0345	0.0012	***1.5260***	***0.6440***	***5.6150***	***0.0180***	***4.6020***

Statistically significant results are bold-enhanced and italic.

## Data Availability

The data presented in this study are available in a Appendix A.

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
