# Peer review of "Reliability of a Risk-Factor Questionnaire for Osteoporosis: A Primary Care Survey Study with Dual Energy X-ray Absorptiometry Ground Truth"

_ijerph, 2021, doi:10.3390/ijerph18031136_

Round 1

Reviewer 1 Report

In this work the authors perform a retrospective study of patients to demonstrate low reliability of a standard osteoporosis risk factor questionnaire. 

This manuscript is clear and straight to the point. Nevertheless two main points, regarding novelty and significance of the content, should be addressed before publication:

1) The authors are questioning the reliability of a standard questionnaire they prepared on the basis of NOF criteria as well as the recommendations of the American College of Radiology (ACR), the ISCD, and the IAEA. The main conclusion they draw from their study is that their own questionnaire is not a reliable tool for the prediction of osteoporosis. The authors propose a targeted physician-patient interaction and reliable risk factor estimation.  Isn't that obvious? Where is the novelty? Nobody is expecting to draw any conclusion from a survey. Please make an effort to address this point. 

2) If the questionnaire is not reliable, where are the areas of improvement? What are the proposed risk factors that should be introduced? How do the authors think they could increase the patients self-awareness and lower their uncertainty?

Author Response

Reviewer 1

In this work, the authors perform a retrospective study of patients to demonstrate the low reliability of a standard osteoporosis risk factor questionnaire. 

This manuscript is clear and straight to the point. Nevertheless, two main points, regarding novelty and significance of the content, should be addressed before publication:

RE: We thank the reviewer for devoting his/her time to reading our manuscript despite the corona-related schedule restrictions. We improved the manuscript after your peer comments and addressed the raised points in the structured response below.  

R.1.1. The authors are questioning the reliability of a standard questionnaire they prepared based on NOF criteria and the recommendations of the American College of Radiology (ACR), the ISCD, and the IAEA. The main conclusion they draw from their study is that their own questionnaire is not a reliable tool for the prediction of osteoporosis. The authors propose a targeted physician-patient interaction and reliable risk factor estimation. Isn't that obvious? Where is the novelty? Nobody is expecting to draw any conclusion from a survey. Please make an effort to address this point. 

RE: This is a fair point. Unfortunately, in daily clinical care, it is not obvious that the survey is filled with a physician's assistance. Surveys are usually filled-in by patients in the waiting room without supervision, and those data are fed into the FRAX calculator by the radiographer-in-charge. We modified the discussion to highlight this information:

"…Even in financially strong, adequately equipped radiological departments, the standard-of-care osteological imaging is performed by radiographers with remote medical supervision. A questionnaire including FRAX-related factors, such as the one tested in this study, is filled by the patients without one-to-one medical support. The risk factors are subsequently fed in the FRAX-calculator to estimate the FRAX-score and determine the need-to-treat according to the 10-year fracture risk (WWW.SHEFFIELD.AC.UK/FRAX/TOOL). The beneficial implications of obtaining a valid patient questionnaire are obvious since anamnestic accuracy influences the treatment decision of osteoporosis."

Moreover, despite the reviewer's justified and fair protesting (and the authors of this manuscript), "drawing conclusions out of a survey" is suggested as a cost-effective method to screen for osteoporosis under conditions of low DXA-unit availability, indeed. The authors of this publication entirely agree with the reviewer that this should not be possible.

"…. non-DXA screening protocols, such as the OP-screening with a questionnaire and quantitative ultrasound [45] or using the Osteoporosis Screening Tool (OST) [46,47]. THE EVIDENCE-BASED SCREENING FOR OP IS APPROACHED BY THE RISK-STRATIFIED OSTEOPOROSIS STRATEGY EVALUATION (ROSE) STUDY [48,49], WHICH IMPLEMENTS A TWO-STEP PROCESS WITH SURVEY SCREENING AND DXA SCREENING OF A SELECTED POPULATION. The current study reveals that implementation of questionnaires as a standalone or first-line screening method calls for cautious data interpretation and requires critical medical supervision to provide reliable results." 

R.1.2. If the questionnaire is not reliable, where are the areas of improvement? What are the proposed risk factors that should be introduced? How do the authors think they could increase the patient's self-awareness and lower their uncertainty?

RE: We modified the conclusions to highlight the suggested improvement areas

"The NOF-questionnaire is a determinator of the FRAX-index and tunes therapeutic decisions. This study tests the reliability of an extended NOF-questionnaire version as applied in the osteological outpatient clinic.  Summarizing the findings, from 24 questioned OP-related risk factors, only eating disorders (OR 10.36) and cancer (OR 4.6) were associated with an increased OP incidence.  Many established OP risk factors, including numerous FRAX-estimates, were not associated or even negatively associated with our study's OP-occurrence. Low self-awareness and patient uncertainty are considered major confounding factors. This result puts in doubt the reliability of patient surveys as stand-alone tools for the prediction of OP. We believe that a targeted physician-patient interaction and reliable risk factor estimation are necessary areas of improvement and in the outpatient osteoporosis care."       

Reviewer 2 Report

Dual-energy x-ray absorptiometry (DXA) that examines bone mineral density at the spine and hip is used to diagnose osteoporosis, to assess patients' risk of fracture, and to monitor response to treatment. In the present study, the authors aim to define the reliability of risk-factors questionnaires collected for 553 patients in 3 years. The results are based on 24 questioned OP-related risk factors. Only eating disorders and cancer have been associated with an increased OP incidence. The manuscript is well organized and clearly written. The tables and figures are of good quality and complete the text adequately. Although the discussion appropriately addresses the topic it might be extended. In my opinion, the manuscript is suitable for publication in this Journal after minor revision from the authors.

Author Response

Reviewer 2

Dual-energy x-ray absorptiometry (DXA) that examines bone mineral density at the spine and hip is used to diagnose osteoporosis, assess patients' risk of fracture, and monitor treatment response. In the present study, the authors aim to define the reliability of risk-factors questionnaires collected for 553 patients in 3 years. The results are based on 24 questioned OP-related risk factors. Only eating disorders and cancer have been associated with an increased OP incidence. The manuscript is well organized and clearly written. The tables and figures are of good quality and complete the text adequately.

R.2.1. Although the discussion appropriately addresses the topic, it might be extended. In my opinion, the manuscript is suitable for publication in this Journal after minor revision from the authors.

RE: We thank the reviewer for devoting his/her time to reading our manuscript despite the corona-related time restrictions. We improved the manuscript after your peer comments and extended the discussion, especially regarding the topic of the unsupervised questionnaire filling, which likely allows for uncertain responses. All text chances are marked using the review tracking tool in MS-word.   

"This study is directly intercalated to the quality of daily clinical management. Even in a financially strong, adequately equipped radiological department, the standard-of-care osteological imaging is performed by radiographers with remote medical supervision. A questionnaire including FRAX-related factors, such as the one tested in this study, is filled by the patients without one-to-one medical support. The risk factors are subsequently fed in the FRAX-calculator to estimate the FRAX-score and determine the need-to-treat according to the 10-year fracture risk (www.sheffield.ac.uk/FRAX/tool). The beneficial implications of obtaining a valid patient questionnaire are obvious since anamnestic accuracy influences the treatment decision or osteoporosis. Moreover, the accuracy and reliability of the risk factor survey is vital for high-risk patient selection, especially in non-DXA screening protocols, such as the OP-screening with a questionnaire and quantitative ultrasound [45] or using the Osteoporosis Screening Tool (OST) [46,47]. The evidence-based screening for OP is approached by the Risk-stratified Osteoporosis Strategy Evaluation (ROSE) study [48,49], which implements a two-step process with survey screening and DXA screening of a selected population. The current study reveals that implementation of questionnaires as a standalone or first-line screening method calls for cautious data interpretation and requires critical medical supervision to provide reliable results."

Reviewer 3 Report

the authors presented an interesting study, however I have some minor methodological concerns:

1) an English check is mandatory from a Native English speaker to improve clarity of text

2) line 85-88 should be moved at the beginning of the results section

3) line 171-172 please provide IRB protocol number

Author Response

Reviewer 3

the authors presented an interesting study; however, I have some minor methodological concerns:

RE: We thank the reviewer for devoting his/her time to reading our manuscript despite the corona-related time restrictions. We improved the manuscript after your peer comments and addressed the raised points in the structured response below.   

R.3.1. an English check is mandatory from a Native English speaker to improve the clarity of text

RE: A professional language proof was performed, and the receipt is attached as an additional document

R.3.2. line 85-88 should be moved at the beginning of the results section

RE: corrected

R.3.3. line 171-172, please provide IRB protocol number

RE: The IRB number is now added in the ethical approval statement, and the IRB permission is uploaded as an original file for the editor's interest.

Reviewer 4 Report

I was looking for more information about surveying, how FRAX risk factors were established, their importance, need for surveying in diagnosis, history etc. It is unclear how the questions in this survey differed (if at all) from those used to establish OP risk factors. Otherwise the paper is very well written and described. 

Author Response

Reviewer 4

R.4.1. I was looking for more information about surveying, how FRAX risk factors were established, their importance, the need for surveying in diagnosis, history, etc. It is unclear how the questions in this survey differed (if at all) from those used to establish OP risk factors. Otherwise, the paper is very well written and described. 

RE: We thank the reviewer for devoting his/her time to reading our manuscript despite the corona-related time restrictions. We improved the manuscript after your peer comments and extended the introduction to describe the questionnaire content in the methods part 2.2

"The questionnaire (Table S1) was designed by AM based on the clinical factors predisposing to osteoporosis as defined by the NOF criteria [3] as well as the recommendations of the American College of Radiology (ACR), the ISCD, and the IAEA [11,26]. Apart from the FRAX-determining factors (previous fracture, fracture in family history smoking, glucocorticosteroids, rheumatoid arthritis, secondary osteoporosis, and alcohol consumption), the questionnaire applied in this study is enriched in risk factors statistically correlated with osteoporosis such as eating disorders and hormonal dysfunction [27]."     

Reviewer 5 Report

Comments to the paper "Reliability of a risk-factor questionnaire for osteoporosis: a primary care survey study with Dual Energy X-ray Absorptiometry ground truth" by Radeva et al.

The authors of this paper performed a retrospective study comparing the cases of osteoporosis observed from analytical instrumentation with a survey regarding their clinical parameters obtained from questionnaires. The paper is generally well written, the aim of the study is clear, methods are adequately organised and results are clearly presented. There are some minor issues to be revised in my opinion.

Line 51: Please define NOF here, instead of in line 60.

Line 79: please remove parenthesis in figure 1.

Lines 127-129: Please insert references of other studies using T-scores and Z-scores.

Line 181: please remove parenthesis in table S2.

Line 186: I would suggest to change section 3.2 title, since it does not describe the actual content of the paragraph.

Line 201: please remove m/σ symbol, since it can be confused with mean/standard deviation ratio.

Lines 203-206: The data discussed in caption should be moved in the main text of results.

Figure 3c-e: Please indicate the meaning of the black diagonal line. Is it the average value or the median?

Line 356: I would rephrase the first sentence as "Nonetheless, the limited number of samples hampered some possible correlations observed from the survey".

Lines 356-360: Are there studies which found positive correlations with these clinical issues and osteoporosis? the authors should compare their findings with other studies in this case

Author Response

Reviewer 5

Comments to the paper "Reliability of a risk-factor questionnaire for osteoporosis: a primary care survey study with Dual Energy X-ray Absorptiometry ground truth" by Radeva et al.

The authors of this paper performed a retrospective study comparing the cases of osteoporosis observed from analytical instrumentation with a survey regarding their clinical parameters obtained from questionnaires. The paper is generally well written, the aim of the study is clear, the methods are adequately organized, and the results are clearly presented. There are some minor issues to be revised, in my opinion.

RE: We thank the reviewer for devoting his/her time to reading our manuscript despite the corona-related time restrictions. We improved the manuscript after your peer comments and addressed the raised points in the structured response below.  

R.5.1. Line 51: Please define NOF here, instead of in line 60.

RE: corrected, thanks for noticing!

R.5.2. Line 79: please remove the parenthesis in figure 1.

RE: corrected

R.5.3. Lines 127-129: Please insert references to other studies using T-scores and Z-scores.

RE: 2.3 L157. "Excellent literature on the technical aspects and clinical applications of the T- and Z-score is reviewed by the Human Health Series of IAEA [29]

R.5.4. Line 181: please remove parenthesis in table S2.

RE: corrected

R.5.5. Line 186: I would suggest changing section 3.2 title since it does not describe the actual content of the paragraph.

RE: Subtitle 3.2 was modified to "The osteoporosis incidence is affected by the Bone Mineral Density measuring site" to better describe the data.

R.5.6. Line 201: please remove the m/σ symbol, since it can be confused with the mean/standard deviation ratio.

RE: corrected

R.5.7. Lines 203-206: The data discussed in the caption should be moved to the main text of the results.

RE: Corrected. The caption of Fig.2 a,b,c,d is now reflected in the results' section 3.1 and 3.2. Thanks for critically reading the results part conjugated to the captions.  

R.5.8. Figure 3c-e: Please indicate the meaning of the black diagonal line. Is it the average value or the median?

RE: Thanks for noticing this unclear point. The solid line represents the correlation coefficient of Spearman (rs). This was added in Fig. 3, along with the original values of rs for each paradigm.

R.5.9. Line 356: I would rephrase the first sentence as "Nonetheless, the limited number of samples hampered some possible correlations observed from the survey."

RE: Line 440: "Nonetheless, the limited number of samples hampered some possible correlations observed from the survey such as ..."

R.5.10. Lines 356-360: Are there studies that found positive correlations between these clinical issues and osteoporosis? the authors should compare their findings with other studies in this case

RE: L 444 "Positive correlations of those factors with osteoporosis are competently reviewed in high reputation specialty books [27,53]. "

Round 2

Reviewer 1 Report

The authors have replied very well to both my points and I am satisfied with the improvements they provided because the manuscript can now be read and appreciated also by a wider audience.